# Discrepancy of C-Reactive Protein, Procalcitonin and Interleukin-6 at Hospitalization: Infection in Patients with Normal C-Reactive Protein, Procalcitonin and High Interleukin-6 Values

**DOI:** 10.3390/jcm11247324

**Published:** 2022-12-09

**Authors:** Eun-Hwa Lee, Kyoung-Hwa Lee, Young-Goo Song, Sang-Hoon Han

**Affiliations:** Division of Infectious Disease, Department of Internal Medicine, Yonsei University College of Medicine, Seoul 06273, Republic of Korea

**Keywords:** C-reactive protein, discrepancy, infection, inflammation, interleukin-6, procalcitonin

## Abstract

C-reactive protein (CRP) or procalcitonin (PCT) alone has limitations in the early detection of infection or inflammation due to shortcomings in specificity and varied cut-off values. Recently, interleukin (IL)-6 has been assessed, but it is not known to what extent the three values are homogeneous in reality. This retrospective study was conducted with two large datasets (discrepancy set with results within 24 h of admission [7149 patients] and follow-up set until 2 weeks of hospital stay [5261 tests]) consisting of simultaneous examinations of CRP, PCT, and IL-6 between January 2015 and August 2021. The specific discrepant group (*n* = 102, 1.4%) with normal CRP (<10 mg/L) and PCT (<0.1 ng/mL) and high IL-6 (≥100 pg/mL) values was extracted from the discrepancy set. Dimensionality reduction and visualization were performed using Python. The three markers were not clearly clustered after t-distributed stochastic neighbor embedding. Pearson’s correlation coefficients between two markers were substantially low (0.23–0.55). Among the high normalized IL-6 levels (≥0.5) (*n* = 349), 17.8% and 38.7% of CRP and PCT levels were very low (≤0.01). 9.2% and 13.4% of normal CRP (*n* = 1522) had high PCT (≥0.5 ng/mL) and IL-6 (≥100 pg/mL) values, respectively. Infection and bacteremia among 102 patients occurred in 36 (35.3%) and 9 (8.8%) patients, respectively. In patients with bacteremia, IL-6 was the first to increase, followed by PCT and CRP. Our study revealed that CRP, PCT, and IL-6 levels were considerably discrepant, which could be misinterpreted if only CRP tests are performed.

## 1. Introduction

Rapidly and properly distinguishing the presence of infection including suspicious or subclinical presentations at the time of hospitalization has clinical significance in two respects: (1) early antimicrobial treatment in infectious diseases can reduce mortality and morbidity, (2) if not suspected, inappropriate antibiotics can be reasonably avoided, which helps to implement antibiotic stewardship [1,2,3]. However, it may not be easy to correctly discriminate between infectious and non-infectious inflammation in relation to immune response that could be manifested by similar symptoms and signs, such as fever, myalgia and alteration of vital signs, or unremarkable appearance [4]. Although many researchers have worked to discover ideal biomarkers that can aid early detection of infection and guidance to antimicrobial therapy, as well as evaluation of inflammation severity and prediction of prognosis in various conditions including sepsis syndrome, autoimmune diseases, trauma, cancer, post-operative period, or ischemia [5,6,7,8,9], C-reactive protein (CRP), procalcitonin (PCT), and interleukin (IL)-6 are the inflammatory markers that are widely being performed in clinical practice outside the realm of research [1,2,5,10,11].

The major problem with many inflammatory markers being explored is that their sensitivity and specificity to identify infection or sepsis vary, and cut-off values applied in the studies are diverse and not validated [5,12,13]. CRP or high-sensitivity CRP has the greatest sensitivity to any type of inflammation, including anaphylactic shock and acute pancreatitis; however, its specificity is low for infection [1,4,12,13,14,15]. Although PCT had been considered to have promising tools for early diagnosis of infection, which has better specificity and negative predictive value (NPV) than CRP [16,17,18], it is now clear that the diagnostic power of PCT in adults is limited except its role in guiding antibiotic stewardship (discontinuation rather than initiation) [18,19,20,21,22,23,24]. Moreover, a suitable and uniform cut-off level for PCT to exclude bacteremia and sepsis has not been suggested [4,24]. Elevations in IL-6 values could reflect the critical and exaggerated immune responses, which are also referred to as a cytokine storm [25]. However, the pleotropic effects and the volatility over time of IL-6 make it intricate for blood concentration measurements to predict infection and determine prognosis [26,27,28]. In addition, IL-6 may be a biomarker for a variety of diseases such as atrial fibrillation, asthma, or malignancies [29,30,31].

To overcome these drawbacks, testing for single inflammatory biomarkers without optimal combinations will increase the likelihood of misinterpretation of infection, which could lead to an irreducible delay in timely antimicrobial treatment and poor outcome or, conversely, overuse of antibiotics [4,24]. Physicians tend to rule out infection in patients with normal CRP and PCT levels and ambiguous clinical presentations. This may be reasonable considering the high sensitivity of CRP and PCT, and the high NPV of PCT [4]. However, there is often a large difference between CRP and PCT values that are simultaneously performed depending on the time from the onset of inflammation to the execution of the examination or various factors [4,5]. In such cases, it could be confused as to which value should be considered more important and how to interpret it, and whether to administer antibiotics or not.

Although it is helpful to examine several inflammatory biomarkers together and to interpret their results appropriately in differentiating infectious and non-infectious inflammation, the heterogeneity of CRP, PCT, and IL-6 levels measured at the same time and their alteration patterns have not yet been analyzed. In this study, we analyzed how often the simultaneous tests for CRP, PCT, and IL-6 on the same day have large diversity, how they change over time, and the characteristics of patients with normal CRP and PCT levels and high IL-6 levels at admission, which may lead to erroneous clinical judgments of suspected infection.

## 2. Materials and Methods

### 2.1. Study Design and Data Preprocessing

We retrospectively retrieved all results of CRP, PCT, and IL-6 measured with the same peripheral blood on the same date among hospitalized patients 18 years of age and older between January 2015 and August 2021 using the query-based relational database management system (RDBMS) of the Yonsei University Health Center in the Gangnam Severance Hospital, a tertiary-care university-affiliated teaching hospital. Among the 14,111 concurrent examinations from 8729 hospitalizations of 7153 patients without severe liver disease of Child-Pugh Score 3 or recent IL-6 inhibitor therapy within 3 months, we included 7149 tests from 7149 patients in the study after removing tests performed 24 h after admission and at re-admission on the same patient (discrepancy dataset). We additionally collected the follow-up 5261 simultaneous tests performed within 2 weeks of admission from 1945 hospitalizations of 1827 patients to evaluate the dynamic changes in each biomarker according to the hospital day (HD) (follow-up dataset) (Figure 1). Date on date of death and last visit (30-day all-cause crude mortality), bacteremia, or fungemia occurring within 3 days of hospitalization, emergency room (ER) visits, and main reasons for admission were also retrieved from the same RDBMS. All biomarker tests were performed based on the clinical need of the medical staff to determine the presence and severity of infection or inflammation according to the presentations and vital signs. Our dataset did not include the tests of high-sensitivity CRP.

Results displayed above the maximum or below the minimum in both datasets were trimmed to the maximum and minimum values, respectively (Appendix A). To determine the overall and mutual distribution of the three markers with different units and normal ranges, all data were transformed to a value between 0 and 1 through the normalization process by maximum and minimum values using the scikit-learn library (version 1.1.2), which is a machine-learning framework of the Python language (version 3.10.6). The normalized data were plotted in a three-dimensional space using the matplotlib visualization library (version 3.5.3) of Python to check the CRP, PCT, and IL-6 points at a glance (Figure 1). Then, we performed dimensionality reduction into two dimensions to determine if the normalized data can be clustered into categories through t-distributed stochastic neighbor embedding (t-SNE) algorithms by scikit-learn. To examine the mutual concordance of the two marker pairs among CRP, PCT, and IL-6, we created the heatmap graphs for total normalized values or counts belonging to each equal interval (ten sections for CRP and IL-6 and eight for PCT) using the seaborn visualization library (version 0.12.0) of Python.

All processes of extraction and preprocessing of the measured values and clinical information were performed with anonymized and randomly numbered data. For this reason, our institutional review board approved this study without the patient’s informed consent (approval No: 3-2022-0318).

### 2.2. Group Selection and Data Collection

From the data-mining process with t-SNE results and the clinical judgment that discrepancy of three markers could have the most meaningfulness, we pulled out the group comprising patients with normal CRP (<10 mg/L) and PCT (<0.1 ng/mL) but high IL-6 (≥100 pg/mL) (*n* = 102) (Figure 1). We collected the clinical data in this group, including baseline characteristics, co-morbidities, symptoms or signs, presence of infection or bacteremia within 3 days of admission, early antimicrobial treatment, and HD or in-hospital all-cause mortality, through a review of the electronic medical records. The quick sepsis-related organ failure assessment (qSOFA) score was calculated based on the mental status and vital signs [32].

### 2.3. Definitions

Infection was limited to a narrow meaning when bacteria or fungi were identified in sterile bodily fluids or when a special radiologist clearly observed the findings of infection in an imaging examination. Because this study aimed to outline the pitfalls of inflammatory biomarkers, we did not judge infection merely as an increase in CRP, PCT, or IL-6 along with changes in vital signs, including fever. Clinically significant bacteremia or fungemia was defined as the presence of ≥ one positive blood culture within 3 days of admission in patients with infection, as defined above.

### 2.4. Measurements of Inflammatory Markers

CRP (Beckman Coulter^®^ AU5822, Brea, CA, USA) and PCT (VITROS^®^ BRAHMS assay, Ortho Clinical Diagnostics, Raritan, NJ, USA) levels were measured using the immunoassay. IL-6 levels were measured by electrochemiluminescence immunoassay using the Elecsys^®^ assay in Cobas^®^ and 411 analyzers (Roche Diagnostics, Basel, Switzerland). Normal reference ranges were <10 mg/L, <0.1 ng/mL, and <7 pg/mL for CRP, PCT, and IL-6, respectively.

### 2.5. Statistical Analyses

The data were expressed as numbers (percentages), means ± standard deviations, or medians (interquartile ranges). We used the independent Student’s t-test or non-parametric Mann–Whitney U test depending on whether the variable had a normal distribution between the two groups. Categorical variables were compared between the two groups using the chi-square test or Fisher’s exact test. To identify the independent factors associated with infection in the specific group, we performed multivariate logistic regression using a backward elimination selection method based on the probability of the Wald statistic (0.10 of probability step selection) with the variables showing significant differences in univariate analysis. Statistical significance was defined as a two-tailed *p*-value ≤ 0.05. All statistical analyses and visualization, including follow-up values, were performed using the SPSS program (version 25.0; IBM SPSS Statistics, Armonk, North Castle, NY, USA) and the matplotlib or seaborn library of Python, respectively.

## 3. Results

### 3.1. Characteristics of the Discrepancy and Follow-Up Dataset

In the discrepancy dataset, all (6725, 94.1%) but 424 patients were hospitalized after undergoing simultaneous CRP, PCT, and IL-6 tests in the ER. A total of 597 (8.4%) and 672 (9.4%) patients had bacteremia or fungemia, respectively, within 3 days of admission and died within 30 days of admission. The most common initial presentation was fever or chills (36.5%), followed by general weakness or poor oral intake (18.6%) and dyspnea or shortness of breath (13.4%). The follow-up dataset had higher 30-day all-cause in-hospital mortality and rates of bacteremia or fungemia (15.6% and 14.7%, respectively) than the discrepancy dataset (Appendix A). Both datasets did not include confirmed cases of COVID-19.

### 3.2. Concordance of CRP, Procalcitonin and IL-6 in Discrepancy Dataset

The three-dimensional plot revealed that the normalized CRP, PCT, and IL-6 values were densely distributed near zero, without showing a unique correlation or pattern (Figure 1). In 15.1% (1082 of 7149) of the simultaneous tests, all normalized CRP, PCT, and IL-6 corresponded to the range of ≤0.01. However, among the very high IL-6 levels with normalized values of >0.5 (4.9%, 349 of 7149), 38.7% and 17.8% of PCT and CRP tests were included in the range of ≤0.01, respectively. Cases with very high CRP levels (>0.5, 109 of 7149, 1.5%) but low IL-6 and PCT levels (≤0.01) were 2.7% and 32.1%, respectively. 

Data clustering was not observed in the visualization after dimensionality reduction to two using the t-SNE method (Figure 1). In addition, paired plots of the normalized values did not show a significant regression line or association between the two markers (Appendix A). After segmenting the three inflammatory markers, the degree of agreement between each range was evaluated using a heatmap. Approximately 9.2% and 13.4% of the simultaneous tests with completely normal CRP levels (0–10 mg/L) (*n* = 1522) had high PCT (≥0.5 ng/mL) and IL-6 (≥100 pg/mL) values, respectively (red boxes in Figure 2A,B). Among the low PCT values (<0.5 ng/mL) (*n* = 4785), a condition not considered sepsis or critical infection, 19.8% had high IL-6 levels (≥100 pg/mL) (blue boxes in Figure 2C). In the correlation analysis, which contained the normalized inflammatory marker values themselves as well as the markers belonging to the same interval, all Pearson’s correlation coefficients were substantially low (from 0.23 to 0.55) (Figure 2D).

### 3.3. Characteristics of Patients with Normal CRP and PCT Levels but High IL-6 Levels According to the Occurrence of Infection

Among the 102 patients with completely normal CRP and PCT levels but high IL-6 levels (≥100 pg/mL), infectious diseases and bacteremia or fungemia occurred in 36 (35.3%) and 9 (8.8%) patients, respectively, until discharge (median HD of 5 days). Malignancies with or without antineoplastic therapy (31.8%, 21 of 66) were the most common main cause of admission in patients without infection, followed by cardiogenic shock (25.8%, 17 of 66) and post-invasive procedures (15.2%, 10 of 66). The frequency of diabetes mellitus in patients with infection was significantly higher than that in patients without infection (27.8% vs. 12.1%, *p* = 0.047). Age, male sex, Charlson comorbidity index, and qSOFA score were not different between patients with and without infection. Patients with infection had a significantly higher frequency of fever (77.8% vs. 51.5%, *p* = 0.011) and IL-6 levels (1450 ± 791 vs. 674 ± 149 pg/mL, *p* = 0.007) at admission than those without infection, but significantly lower serum creatinine levels (0.9 ± 0.5 vs. 1.5 ± 1.1, *p* = 0.031). Regardless of the presence of infectious diseases, the majority of patients in this group received antibiotics at the beginning of hospitalization at a similarly high rate (64–78%). ICU treatment history (19% vs. 30%) and all-cause mortality (19% vs. 21%) were similar between patients with and without infection, but the rates were considerably higher (Table 1). Fever at admission was the only independent clinical factor associated with infection (OR 2.7; 95% CI:1.0–7.1, *p* = 0.043) (Table 2).

### 3.4. Changing Patterns of Concurrent CRP, Procalcitonin, and IL-6 Levels

The patient group from the discrepancy dataset showed that CRP and PCT levels, which were normal on the day of admission, increased significantly 24 h after admission. In patients with infection, the CRP level decreased with the time of hospitalization, but in patients without infection, the CRP level remained high. Notably, the CRP values did not differ between the infected and non-infected groups on the first and second days after admission. PCT levels showed maximum values on the first and second day of hospitalization in patients with and without infection, respectively. In patients with infection, PCT levels decreased steadily over time. Even in the case of IL-6, it was observed that IL-6 decreased to a very low value after 3 days of hospitalization in the presence of infection, but increased with severe changes in the absence of infection (Appendix A).

Patients with bacteremia or fungemia from the follow-up dataset had the highest mean CRP and PCT levels on the first day of admission. IL-6 elevation was observed before the highest CRP and PCT levels (Figure 3A). No steady decrease in CRP, PCT, and IL-6 levels was observed in the absence of bacteremia. Regardless of the occurrence of bacteremia or fungemia, the CRP, PCT, and IL-6 levels showed significant differences on each day of hospitalization, and this discrepancy was larger in the absence of bacteremia (Figure 3A,B). CRP and IL-6 levels continued to increase until 14 days after admission in patients who died within 30 days, which reflects more severe infection or non-infectious inflammatory conditions. In patients who survived for up to 30 days, CRP values continued to be low after 3 days of hospitalization (Figure 4).

## 4. Discussion

Our study showed that the values of inflammatory biomarkers in the simultaneous test were substantially different and uncorrelated. These results suggest that infection could be misinterpreted, especially underdiagnosed, in the early stage of admission if only one test is performed or if short-term follow-up examinations are not adequately performed. Considering the special group showing the extreme discrepancy we found (normal CRP and PCT, but high IL-6, and not low frequency of 1.4% in a large dataset), there would be some cases in which infection or inflammation may be incorrectly determined or may not be detected very early following admission, even with a combination test of CRP and PCT, because IL-6 is a less frequently used biomarker in current clinical settings.

Interestingly, very high PCT or IL-6 levels with low CRP levels were more frequently observed than high CRP and low PCT or IL-6 levels (see the heatmaps in Figure 2). Based on these findings, it could be inferred that the CRP test alone may increase the risk of missing early susceptibility to critical infection or inflammation. Given that the clinical symptoms or signs may not be evident even in sepsis, particularly old age or early visits, normal or low levels of initial CRP without simultaneous measurement of other biomarkers could allow late administration of antibiotics and lead to poor clinical outcomes. 

Because the featured group in this study could often be unexpected and seldom encountered in real clinical practice, their characteristics, with normal CRP and PCT but high IL-6, will give us implications for the necessity for initial IL-6 measurement. Only one-quarter of the patients presented with both fever and shock, suggestive of severe inflammatory conditions. However, the fact that the ICU treatment history and mortality rate were relatively high despite a short hospital stay may indicate that the patients in this group were already in a hazardous medical situation, including cytokine storms at admission. Although both CRP and PCT levels were normal, numerous patients were able to receive antibiotics early in their hospital stay, probably because of the high IL-6 value. However, higher IL-6 levels were not independently associated with early infection in this group. Taken together, it is assumed that IL-6, similar to CRP [1,15], would comprehensively represent inflammation status caused by various causes rather than the infection itself [33,34].

Because CRP or PCT alone has limitations as a diagnostic purposes of infection (particularly its severity) [1,4,5,18,24], it would be helpful to examine the combined inflammatory biomarkers while considering diagnostic stewardship with avoidance of over-tests [35]. The discrepancy from our study suggests that measuring IL-6 with CRP and/or PCT may be useful to detect infection or bacteremia more quickly and correctly in diverse situations [27,34,36,37]. Physiologically, CRP is synthesized in the liver in response to stimulation of IL-6 increased by inflammation including infection [13,15]. Although blood IL-6 level can be rapidly reduced [27], IL-6 could be particularly helpful in early diagnosis because IL-6 is produced before CRP [13,15,34,37]. However, we must prudently decide the antimicrobial treatment to avoid overuse of antibiotics, because a single increase in IL-6 does not directly indicate infection or bacteremia [33,34,38].

Long-term kinetic profiles of patients with early infection or bacteremia from the large follow-up dataset, which is the first report to analyze simultaneous tests from patients instead of healthy volunteers to our knowledge, showed that the order of the date with the maximum value was IL-6 (0 HD) and PCT or CRP (1 HD) [39]. These results also provide additional evidence that IL-6 testing may be helpful immediately after hospitalization. A chronic elevation in IL-6, indicative of a persistent inflammatory state, did not help distinguish a poor prognosis as assessed by 30-day mortality. In contrast, CRP levels remained low in patients without mortality, and continued to increase in patients with poor prognosis.

Our data need to be interpreted carefully, taking into account the time interval between onset of symptoms or signs and first test after the patient’s presentation. This is because the fact that kinetic properties of inflammatory markers after initiation of pathophysiologic alteration is different for each test could affect the more accurate interpretation of the presence and severity of infection or inflammation at the time of examination. Therefore, as the main conclusion of our study, measuring multiple inflammatory markers simultaneously, including IL-6 in particular, rather than one biomarker, will be more helpful in evaluating infectious or inflammatory status properly and in good time. The recently proposed concept of CRP velocity can reflect the dynamic change and may better estimate the inflammatory process than CRP measured only once in a special situation (for example, differentiation between acute bacterial and viral infection and myocardial infarction [40,41,42,43,44].

Numerous previous clinical studies have focused on identifying which inflammatory markers are better predictors by comparatively evaluating the performance (i.e., area under the receiver operative characteristic curve, sensitivity and specificity or adequate cut-off level) in specific situations or populations [5,9,16,18,19,27,28,36,37]. Our study attempted to highlight the pitfalls and caveats of the inflammatory biomarkers by evaluating the heterogeneity in results measured simultaneously with the same blood sample. This study has the following limitations: (1) trimmed data, particularly displayed above the maximum IL-6 level, can affect the distribution of normalized values, (2) the number of follow-up tests was not large enough in a retrospective analysis, and (3) because of possible inaccuracies and missing values in electronic medical recode, our large cohort could not retrospectively collect data on time to symptom or sign onset to apply the dynamic changes of inflammatory markers, (4) the analyses were not conducted on a specific disease group, and (5) it has not been properly evaluated how inappropriate antibiotic administration and inefficient removal of infection source, which are presumed to be very low rate, affect the dynamic change of the inflammatory markers in the follow-up dataset. Nevertheless, our first paper analyzing large-scale data reminds us that caution is needed when interpreting inflammatory markers, and that it is necessary to perform simultaneous tests in combination for a more accurate and faster diagnosis of inflammation including infection. If it is not possible to measure IL-6 levels at the actual clinical setting, we recommend frequent and regular measurements of CRP at short intervals (at least twice a day) without excluding infection prematurely for patients with suspected infection who present with relatively normal or slightly elevated CRP and PCT level.

## 5. Conclusions

This study using a large dataset demonstrated that the concurrent examination of CRP and IL-6 or PCT could be helpful for the early suspicion of infection, but careful interpretation is important because of their discrepancies. In addition, it will be necessary to actively measure IL-6 in actual clinical settings. We need to find new methods and biomarkers to help us with the diagnosis of inflammations caused by microbes or other processes. 

## Figures and Tables

**Figure 1 jcm-11-07324-f001:**
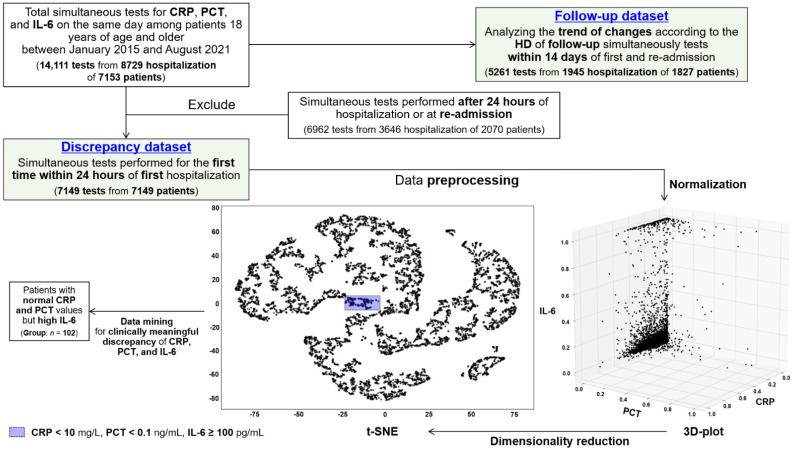
Flow chart showing the process of selecting CRP, PCT and IL-6 tests performed simultaneously (discrepancy and follow-up dataset) and patient groups from data preprocessing and mining. Aberrations: CRP—C-reactive protein; 3D—three dimensional; HD—hospital day; IL—interleukin; PCT—procalcitonin; t-SNE—t-distributed stochastic neighbor embedding.

**Figure 2 jcm-11-07324-f002:**
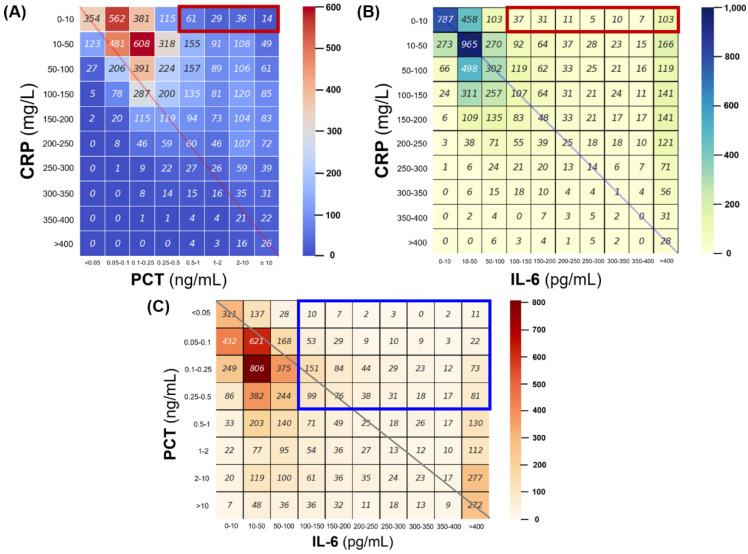
Distribution and correlation between CPR, procalcitonin, and interleukin-6 simultaneously measured at admission. (**A**–**C**) Heatmap for the count of binning range between the three inflammatory markers. The color bar indicates the count of the test numbers in each range. (**D**) Heatmap of Pearson’s correlation coefficient. The color bar indicates Pearson’s correlation coefficients.

**Figure 3 jcm-11-07324-f003:**
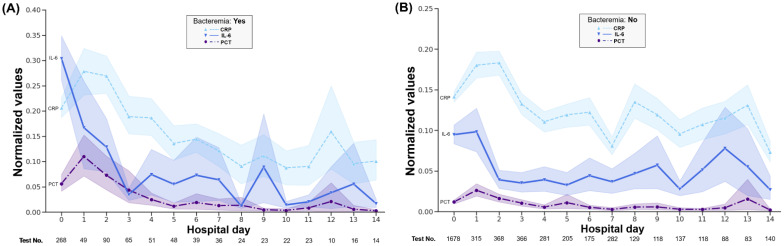
Dynamic change of the normalized CRP, procalcitonin, and IL-6 values in the follow-up dataset according to bacteremia or fungemia (**A**,**B**). Each point represents an average of the measured values. The shading of each line indicates the upper and lower bounds of the 95% confidence interval.

**Figure 4 jcm-11-07324-f004:**
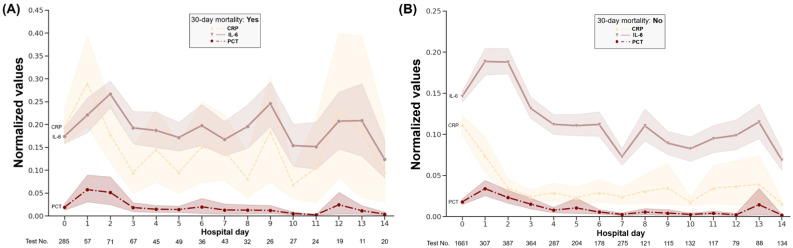
Dynamic change in the normalized CRP, procalcitonin and IL-6 values in the follow-up dataset according to 30-day all-cause in-hospital mortality (**A**,**B**). Each point represents an average of the measured values. The shading of each line indicates the upper and lower bounds of the 95% confidence interval.

**Table 1 jcm-11-07324-t001:** Difference of clinical characteristics and outcomes according to the occurrence of infectious diseases in patients with normal CRP and procalcitonin values but high IL-6 levels in simultaneous tests within 24 h of admission.

Characteristics	Total (*n* = 102)	Infection Diagnosed within 3 Days of Admission		*p*-Values
		Yes (*n* = 36)	No (*n* = 66)	
Age, years	67.1 ± 15.6	69.4 ± 16.5	65.9 ± 15.1	0.281
Sex, male	55 (53.9)	20 (55.6)	35 (53.0)	0.838
BMI, kg/m^2^	23.0 ± 3.9	23.2 ± 3.6	22.8 ± 4.0	0.697
**Underlying co-morbidities** ^a^				
NIDDM	18 (17.6)	10 (27.8)	8 (12.1)	0.047
Chronic heart diseases	6 (5.9)	2 (5.6)	4 (6.1)	0.917
Chronic lung diseases	5 (4.9)	2 (5.6)	3 (4.5)	0.581
Chronic liver diseases	18 (17.6)	5 (13.9)	13 (19.7)	0.590
Chronic renal diseases	6 (5.9)	1 (2.8)	5 (7.6)	0.420
Solid cancers	33 (32.4)	8 (22.2)	25 (37.9)	0.125
Charlson comorbidity index	4 (3–6)	5 (1–7)	4 (3–5)	0.427
Vaccination history ^b^	31 (30.4)	12 (33.3)	19 (28.8)	0.637
**Clinical presentations at admission**				
Fever ^c^	62 (60.8)	28 (77.8)	34 (51.5)	0.011
qSOFA	1 (0–2)	1 (0–2)	1 (0–2)	0.842
Altered mentation, GCS < 15	24 (23.5)	5 (13.9)	19 (28.8)	0.142
RR ≥ 22 breaths/min	23 (22.5)	10 (27.8)	13 (19.7)	0.459
Systolic BP ≤ 100 mmHg	58 (56.9)	21 (58.3)	37 (56.1)	0.664
Fever and shock ^c^	20 (19.6)	8 (22.2)	12 (18.2)	0.795
**Laboratory tests ^d^**				
CRP (mg/L)	3.5 ± 2.7	3.3 ± 2.7	3.9 ± 2.7	0.320
Procalcitonin (ng/mL)	0.15 ± 0.10	0.15 ± 0.11	0.14 ± 0.08	0.441
IL-6 (pg/mL)	947.9 ± 699.6	1449.5 ± 791.2	674.3 ± 149.2	0.007
Creatinine (mg/dL)	1.3 ± 1.0	0.9 ± 0.5	1.5 ± 1.1	0.031
PT, INR	1.4 ± 0.7	1.2 ± 0.5	1.5 ± 0.7	0.054
Bacteremia or fungemia ^e^	9 (8.8)	9 (25.0)	―	―
**Non-infectious conditions**				―
Anaphylaxis or drug-related AR	2 (2.0)	―	2 (3.0)	
Cancer or anti-neoplastic therapy	21 (20.6)	―	21 (31.8)	
Cardiogenic shock	17 (16.7)	―	17 (25.8)	
Chronic organ failure	2 (2.0)	―	2 (3.0)	
CNS diseases ^f^	3 (2.9)	―	3 (4.5)	
Connective tissue disease	1 (1.0)	―	1 (1.5)	
Hypovolemic shock ^g^	7 (6.9)	―	7 (10.6)	
Post-invasive procedures	10 (9.8)	―	10 (15.2)	
Trauma including fracture	3 (2.9)	―	3 (4.5)	
**Antimicrobial therapy**				
Within 24 h of admission	67 (65.7)	25 (69.4)	42 (63.6)	0.664
Within 48 h of admission	71 (69.6)	28 (77.8)	43 (65.2)	0.260
**Outcomes**				
Hospital stays—days	5.0 (1.5–12.5)	7.1 (2.5–14.6)	2.7 (1.5–8.7)	0.023
Invasive MV treatment	31 (30.4)	6 (16.7)	25 (37.9)	0.041
ICU admission—yes	27 (26.5)	7 (19.4)	20 (30.3)	0.253
In-hospital all-cause mortality	21 (20.6)	7 (19.4)	14 (21.2)	0.833

Data are expressed as numbers (percentage), mean ± standard deviation, or median (interquartile range). ^a^ There were no patients with inflammatory bowel disease, solid organ or any hematopoietic stem cell transplant recipients, or patients taking medication that alters the immune response (for example, long-term corticosteroids, monoclonal antibodies, etc.) except one patient with connective tissue disease. ^b^ Within 3 months prior to the first concurrent examination of CRP, PCT, and IL-6. Vaccines included COVID-19, pneumococcus, Influenza, and herpes zoster, etc. ^c^ Fever and shock were defined as a temperature of >38 C or <36·Cand systolic BP ≤ 100 mmHg, respectively. ^d^ First peripheral blood tests within 24 h of admission. ^e^ Means a case in which microorganisms were identified in blood cultures conducted within 3 days of admission. ^f^ Hemorrhage or stroke. ^g^ Including bleeding events. Aberrations: AR—adverse reaction; BMI—body mass index; BP, blood pressure; CNS, central nervous system; CRP, C-reactive protein; GCS, Glasgow coma scale; ICU—intensive care unit; IL—interleukin; INR—international normalized ratio; MV—mechanical ventilation; NIDDM—non-insulin dependent diabetes mellitus; PT—prothrombin time; qSOFA—quick sequential organ failure assessment; RR—respiratory rate.

**Table 2 jcm-11-07324-t002:** Independent factors associated with infectious disease in patients with normal CRP and procalcitonin values but high IL-6 levels within 24 h of admission.

Variables	OR	95% CI	*p*-Values
Underlying diseases—yes			
NIDDM	2.86	0.91–8.97	0.072
Fever at admission—yes	2.70	1.03–7.05	0.043
Laboratory tests within 24 h of admission			
IL-6 > 500 pg/mL ^a^	1.70	0.70–4.12	0.242
Creatinine < 0.9 mg/dL ^a^	0.84	0.34–2.08	0.710

^a^ This group included 49 (48%) and 50 (49%) patients with IL-6 levels > 500 pg/mL and creatinine < 0.9 mg/dL, respectively. Aberrations: CI—confidence interval; CRP—C-reactive protein; IL—interleukin; OR—odds ratio; NIDDM—non-insulin-dependent diabetes mellitus.

## Data Availability

The data are available upon request from the corresponding author.

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
