# Peer review of "Discrepancy of C-Reactive Protein, Procalcitonin and Interleukin-6 at Hospitalization: Infection in Patients with Normal C-Reactive Protein, Procalcitonin and High Interleukin-6 Values"

_jcm, 2022, doi:10.3390/jcm11247324_

Round 1

Reviewer 1 Report

The manuscript is interesting, well written and presents fairly common scenario of the need to early detect infection in order to enable the clinician to make treatment related decisions in a timely manner. As described in the manuscript, inflammatory biomarkers are widely used in order to establish this goal even though a discrepancy may exist between the different biomarkers. The authors discuss biomarker combination to better assess infection.

The author should address the matter of time difference between symptoms onset and patient presentation. For example, as two patients with the same CRP, PCT and IL-6 levels, but one patient has infection lasting one day at presentation and the other has infection lasting for 10 days at presentation, may require different workup and treatment.

The authors themselves address this matter at the introduction section (lines 70-71), but no relevant data regarding the study participants is provided to the reader.

The authors describe the change of biomarkers levels overtime and its prognostic value. Several studies have demonstrated the use of the kinetic properties of inflammatory biomarkers, particularly CRP, to early detect developing infection (I refer the authors to the recently published review of CRP velocity biomarker published by Levinson et al, at the International Journal of Molecular Science). The authors should address this kinetic property of inflammatory biomarker and how it compares to their suggestion to combine different biomarkers to detect infection.

The study took place during the COVID-19 pandemic, however no information is provided about COVID-19. Were any of the participants diagnosed with COVID-19 during the study? Were they excluded? COVID -19 is associated with elevated biomarkers so the issue should be addressed.

Table 1 presents demographic and clinical characteristics of the study participants, however not the table nor the text provide crucial information about the participants, which can effect their clinical presentation and biomarker levels- how many participants had medical history of inflammatory disease (for example: inflammatory bowel disease, rheumatoid arthritis etc.), how many took medications that alter the immune response (for example: glucocorticoids, monoclonal antibodies etc.), were the participants vaccinated according the recommendations (COVID-19, Influenza, Pneumococcus etc.).

A relatively large number of patients had bacteremia or fungemia. Did the patients receive appropriate antimicrobial treatment? Did the patients have any procedures of infection source control? When were the biomarkers measured in relation to these treatments and did it affect the biomarkers level? On the other hand, if appropriate antimicrobial treatment and source control were not performed, it too would be expected to affect the biomarker level.

At the material and methods section, lines 126-128 “if IL-6 is not…” should be moved to the discussion section.

At the discussion section, lines 315-316 regarding the physiology of CRP and IL-6 should be moved to the introduction section.

CRP is a widely used biomarker by clinicians and to a lesser degree is PCT, However test of IL-6 level is not readily available at most medical facilities, therefore what is the authors recommendation for the clinicians who treat patients with suspected infection who present with relatively normal or slightly elevated CRP and PCT level and no available IL-6 test?

Author Response

  1. The author should address the matter of time difference between symptoms onset and patient presentation. For example, as two patients with the same CRP, PCT and IL-6 levels, but one patient has infection lasting one day at presentation and the other has infection lasting for 10 days at presentation, may require different workup and treatment. The authors themselves address this matter at the introduction section (lines 70-71), but no relevant data regarding the study participants is provided to the reader. The authors describe the change of biomarkers levels overtime and its prognostic value. Several studies have demonstrated the use of the kinetic properties of inflammatory biomarkers, particularly CRP, to early detect developing infection (I refer the authors to the recently published review of CRP velocity biomarker published by Levinson et al, at the International Journal of Molecular Science). The authors should address this kinetic property of inflammatory biomarker and how it compares to their suggestion to combine different biomarkers to detect infection.
  • We totally agree with this valuable comment for onset time of symptoms or signs, which may be considered the starting point of an infectious or inflammatory pathophysiology. Therefore, we clearly agree that need to consider the time interval between the first results of the inflammatory markers (CRP/PCT/IL-6 etc.) performed at the patient's first hospital visit and the estimated time of onset of the inflammation.
  • We took a detailed look at the papers recommended by the reviewers, and fully agree with on the concept and clinical usefulness (importance) of CRP velocity (estimated CRP velocity and CRP velocity upon admission) considering time difference between onset of symptoms/signs and the time of presentation.
  • Here are a little of our concern. Information voluntarily reported by the patients or caregivers would also have the disadvantage that it may lack accuracy in some cases. Therefore, estimated CRP velocity based on information from patients' voluntary reports might be somewhat confusing for various reasons including inability to accurately perceive or remember. This will be especially true in severely infectious or inflammatory conditions accompanied by loss of consciousness (altered mental status) or severe change in vital signs (in other words, critically-ill patients).
  • When all these factors are taken into account, it is judged to be more accurate that ultimately, we could not know the exact onset time of infectious or non-infectious inflammation in several cases in spite of importance of time intervals.
  • An important interest for many physicians will be how severe the inflammatory condition is at the time the patient visits the hospital. Through various assessment tools of severity (several scores for sepsis or critically-illness [for example, quickSOFA, NEWS: National Early Warning Score, MEWS: Modified Early Warning Score]), we will be able to provide more appropriate or intensive treatments to the patient and predict the prognosis.
  • Since there may be differences between various inflammatory markers according to the time interval from onset of symptoms/signs to patient presentation, it is judged that the simultaneous and combined tests with several inflammatory markers including IL-6 may further assist in assessing the severity of infectious or non-infectious inflammation severity (or sometimes confirming the absence of an infection or inflammation) at the first visit. This is the main purpose (suggestion) of our study and the message (key points or conclusion) we want to give to our readers.
  • Although we are fully aware of the importance of the above point, it was practically impossible to add new information for onset time of symptoms and signs because our study was analyzed from a large amount of data (7,149 tests). We ask that reviewers fully understand this difficulty.
  • We further discussed these important issues of our study in the revised manuscript as the sentences below. And we added this point to the limitations of the study. Additionally, we newly inserted the references for CRP velocity.
  1. [Discussion: Line 365-376]: Our data needs to be interpreted carefully, taking into account time interval between onset of symptoms or signs and first test after patient's presentation. This is because the fact that kinetic properties of inflammatory markers after initiation of pathophysiologic alteration is different for each test could affect the more accurate interpretation of the presence and severity of infection or inflammation at the time of examination. Therefore, as the main conclusion of our study, measuring multiple inflammatory markers simultaneously including IL-6 in particular, rather than one biomarker, will be more helpful in evaluating infectious or inflammatory status timely and properly. The recently proposed concept of CRP velocity can reflect the dynamic change and may better estimate the inflammatory process than CRP measured only once in special situation (for example, differentiation between acute bacterial and viral infection and myocardial infarction) [40-44].

  1. [Limitation: Line 385-388]: (3) because of possible inaccuracies and missing values in electronic medical recode, our large cohort could not retrospectively collect data on time to symptom or sign onset to apply the dynamic changes of inflammatory markers,
  • New reference:
  1. 40. Coster, D.; Wasserman, A.; Fisher, E.; Rogowski, O.; Zeltser, D.; Shapira, I.; Bernstein, D.; Meilik, A.; Raykhshtat, E.; Halpern, P. et al. Using the kinetics of c-reactive protein response to improve the differential diagnosis between acute bacterial and viral infections. Infection 2020, 48, 241-248.
  2. 41. Bernstein, D.; Coster, D.; Berliner, S.; Shapira, I.; Zeltser, D.; Rogowski, O.; Adler, A.; Halutz, O.; Levinson, T.; Ritter, O. et al. C-reactive protein velocity discriminates between acute viral and bacterial infections in patients who present with relatively low crp concentrations. BMC Infect Dis 2021, 21, 1210.
  3. 42. Holzknecht, M.; Tiller, C.; Reindl, M.; Lechner, I.; Troger, F.; Hosp, M.; Mayr, A.; Brenner, C.; Klug, G.; Bauer, A. et al. C-reactive protein velocity predicts microvascular pathology after acute st-elevation myocardial infarction. Int J Cardiol 2021, 338, 30-36.
  4. 43. Banai, A.; Levit, D.; Morgan, S.; Loewenstein, I.; Merdler, I.; Hochstadt, A.; Szekely, Y.; Topilsky, Y.; Banai, S.; Shacham, Y. Association between c-reactive protein velocity and left ventricular function in patients with st-elevated myocardial infarction. J Clin Med 2022, 11.
  5. 44. Levinson, T.; Wasserman, A. C-reactive protein velocity (crpv) as a new biomarker for the early detection of acute infection/inflammation. Int J Mol Sci 2022, 23.

  1. The study took place during the COVID-19 pandemic; however, no information is provided about COVID-19. Were any of the participants diagnosed with COVID-19 during the study? Were they excluded? COVID -19 is associated with elevated biomarkers so the issue should be addressed.

  • A total of 1,786 COVID-19 patients (confirmed by real-time RT-PCR for SARS-CoV2) visited our hospital during the study period. According to government policy, all patients who were tested for suspected COVID-19 had to remain isolated in a negative pressure room until negative result was confirmed. COVID-19 patients also had to maintain negative pressure isolation for 7 days from the date of confirmation in the mild or moderate cases and 14 days in the severe cases. IL-6 and procalcitonin testing was not performed on COVID-19 patients due to sample collection and processing problems during the quarantine period. For these reasons, this study did not include the confirmed COVID-19 cases in which three inflammatory markers were simultaneously measured.
  • We have added a description of this content to the revised paper.

[Results. 3.1. Characteristics of the discrepancy and follow-up dataset: Line 173-174]: Both datasets did not include confirmed cases of COVID-19.

  1. Table 1 presents demographic and clinical characteristics of the study participants, however not the table nor the text provide crucial information about the participants, which can affect their clinical presentation and biomarker levels- how many participants had medical history of inflammatory disease (for example: inflammatory bowel disease, rheumatoid arthritis etc.), how many took medications that alter the immune response (for example: glucocorticoids, monoclonal antibodies etc.), were the participants vaccinated according the recommendations (COVID-19, Influenza, Pneumococcus etc.).

  • We have already investigated the medical history you mentioned when preparing the paper. There was only one patient with connective tissue disease (autoimmune inflammatory rheumatic disease including rheumatoid arthritis) in group without infection diagnosed within 3 days of admission. This information has already been included in Table 1.
  • Additionally, there was no patients with inflammatory bowel disease and solid organ or hematopoietic stem cell transplant recipient taking immunosuppressants. In addition, there was no patients taking medication that alter the immune response including long-term corticosteroids, monoclonal antibodies, and IL-6 antagonist (inhibitors). Therefore, we did not include these details in Table 1.

: We already mentioned about the IL-6 inhibitor therapy in the original manuscript

[2. Materials and Methods 2. 1. Study design and data preprocessing. Line 90]: Among the 14,111 concurrent examinations from 8,729 hospitalizations of 7,153 patients without severe liver disease of Child-Pugh Score 3 or recent IL-6 inhibitor therapy within 3 months,

: Other medical histories newly described in detail in the footnote of Table 1. (Since there were no relevant patients, details were not described in the text.)

[Footnote of Table 1, Line 279-282]: aThere were no patients with inflammatory bowel disease, solid organ or hematopoietic stem cell transplant recipient, and patients taking medication that alter the immune response (for example, long-term corticosteroids, monoclonal antibodies, etc.) except one patient with connective tissue disease.

  • We newly inserted data on vaccination history (including COVID-19, Influenza, Pneumococcus etc.) within 3 months prior to the first concurrent examination of CRP, PCT, and IL-6. However, because there are various types of vaccines and the vaccination rates for certain vaccines are very low, data on specific vaccines are not included (Vaccines given within 3 months were almost always COVID-19 vaccines). There was no statistical difference in vaccination history between the two groups.

[Footnote of Table 1, Line 282-283]: bWithin 3 months prior to the first concurrent examination of CRP, PCT, and IL-6. Vaccines included COVID-19, pneumococcus, Influenza and herpes zoster, etc.

  1. A relatively large number of patients had bacteremia or fungemia. Did the patients receive appropriate antimicrobial treatment? Did the patients have any procedures of infection source control? When were the biomarkers measured in relation to these treatments and did it affect the biomarkers level? On the other hand, if appropriate antimicrobial treatment and source control were not performed, it too would be expected to affect the biomarker level.

  • We fully agree with the reviewer's comments and totally understand that appropriate antimicrobial treatment and procedures of infection source control performed timely and properly are very important and that such managements can have a significant impact on values of inflammatory markers measured at the same time.
  • Due to the nature of our cohort, which included a large number of patients, it was not possible to check every patient's record in detail. Nevertheless, since appropriate antibiotic treatment and effective elimination of infection source are the core and very fundamental requirements in managements of infectious diseases (as described in general internal medicine textbooks), we think all the physicians in our hospital have carried out these principles perfectly by consultation with infectious disease specialists or mutually cooperative multidisciplinary care as well as faithfully following the (Survival Sepsis Campaign) guidelines that have been continuously revised for a long time. It is difficult to consider that the above treatment process is not properly carried out in substantial cases at our hospital, which is an affiliated hospital of a medical school and is a tertiary general hospital that receives referrals from other hospitals. In order to provide treatment as a tertiary general hospital in South Korea, all appropriate treatment procedures must be inspected and certified at regular intervals by an authoritative government-affiliated institution for patient safety and quality improvement.
  • The discrepancy dataset, which is the core data of our paper, contained the results of inflammatory markers measured within 24 hours of admission. The first peripheral blood tests after admission to the ER or hospital ward are performed within 1 hour (before administration of antibiotics and any further evaluation including CT scan) in almost all cases. Therefore, even if it is assumed that the time is somewhat delayed, there is no possibility of blood collection for measuring inflammatory markers (simultaneous test of CRP, Procalcitonin, IL-6) after 3 hours. These facts indicate that it is clear that the three inflammatory markers measured simultaneously were performed prior to administration of antibiotics and removal of infectious agents. Therefore, the results of the CRP, PCR, IL-6 in the discrepancy dataset are not affected by the appropriate antimicrobial treatment and control (removal) of infection source.
  • In follow-up dataset, values (changes) of three inflammatory markers over time in hospital days can be affected by appropriate antibiotic treatment and elimination of effective infectious agents. As we answered above, it is certain that catheter insertion (ex, abscess drainage, drainage of exudative pleural fluid, and PCN) (sometimes, removal of catheter which causes infection), procedures (for PTBD, PTGBD), local I & D (incision and drainage) and surgeries were timely and properly performed in almost all cases through very organic cooperation of various departments.
  • We think that it may be difficult to define and evaluate appropriate antibiotic use in our large cohort data, which includes patients from various disease populations. However, in our hospital, the use of antibiotics for almost all infectious diseases is administered under the consultation and supervision of an infectious disease specialist in patients with bacteremia or fungemia, and major important antibiotics associated with serious resistance restricted to be prescribed only by infectious disease specialists (in the computerized prescription system). (Furthermore, when bacteremia and fungemia are present, collaboration with an infectious disease department is essential.)
  • When these points are comprehensively and carefully judged, it is judged that it is extremely rare that inappropriate antibiotics that are resistant to antibiotic susceptibility testing are administered to patients with bacteremia or fungemia.
  • We ask the reviewer to consider these facts comprehensively and fully consider the fact that in practice, inappropriate antibiotic administration and ineffective elimination of infection source are only a small percentage of all patients, and do not affect the overall results and conclusions of this study.
  • We added this important point as one of the limitations in the revised paper.

[Discussion: Line 390-393]: (5) It has not been properly evaluated how inappropriate antibiotic administration and inefficient removal of infection source, which are presumed to be very low rate, affect the dynamic change of the inflammatory markers in the follow-up dataset.

  1. 5. At the material and methods section, lines 126-128 “if IL-6 is not…” should be moved to the discussion section.
  • According to this point along with below comment, we moved this sentence to the discussion section in the revised manuscript.

[Discussion: Line 320-323]: Considering the special group showing the extreme discrepancy we found (normal CRP and PCT, but high IL-6, and not low frequency of 1.4% in a large dataset), there would be some cases in which infection or inflammation may be incorrectly determined or may not be detected very early on admission, even with a combination test of CRP and PCT, because IL-6 is a less frequently used biomarker in current clinical settings.

  1. 6. At the discussion section, lines 315-316 regarding the physiology of CRP and IL-6 should be moved to the introduction section.
  • (Line 315-316): “there would be some cases in which infection or inflammation is ignored early in the onset of pathophysiology”: This paragraph does not provide a detailed description of the pathophysiology of CRP and IL-6. It is addressed that there is a possibility that the initial state of infection or inflammation may not be recognized unless several inflammatory markers, including IL-6, are tested together.
  • We think that the word of “physiology” can be confusing. Along with the comment No. 5, we changed the sentence to below in the revised paper.

[Discussion: Line 320-323]: Considering the special group showing the extreme discrepancy we found (normal CRP and PCT, but high IL-6, and not low frequency of 1.4% in a large dataset), there would be some cases in which infection or inflammation may be incorrectly determined or may not be detected very early on admission, even with a combination test of CRP and PCT, because IL-6 is a less frequently used biomarker in current clinical settings.

  1. 7. CRP is a widely used biomarker by clinicians and to a lesser degree is PCT. However, test of IL-6 level is not readily available at most medical facilities, therefore what is the authors recommendation for the clinicians who treat patients with suspected infection who present with relatively normal or slightly elevated CRP and PCT level and no available IL-6 test?

  • We appreciate the reviewer's valuable comments. In the original paper, we considered this point important and have already described the sentence below.

[Discussion: Line 324]: Considering the special group showing the extreme discrepancy we found (normal CRP and PCT, but high IL-6, and not low frequency of 1.4% in a large dataset), there would be some cases in which infection or inflammation may be incorrectly determined or may not be detected very early on admission, even with a combination test of CRP and PCT, because IL-6 is a less frequently used biomarker in current clinical settings.

[Discussion: Line 353-354]: IL-6 could be particularly helpful in early diagnosis because IL-6 is produced before CRP [13,15,34,37].

  • If it is not possible to measure IL-6 levels at the actual clinical setting, we recommend frequent and regular measurements of CRP at short intervals (at least twice a day) without excluding infection prematurely for patients with suspected infection who present with relatively normal or slightly elevated CRP and PCT level. This is considered equivalent to "CRP velocity (CRPv) upon admission" described by Tal Levinson et al. (Int. J. Mol. Sci. 2022, 23, 8100. https://doi.org/10.3390/ijms23158100).
  • We added the below sentence to the end of the discussion section of the revised manuscript.

[Discussion: Line 396-400]: If it is not possible to measure IL-6 levels at the actual clinical setting, we recommend frequent and regular measurements of CRP at short intervals (at least twice a day) without excluding infection prematurely for patients with suspected infection who present with relatively normal or slightly elevated CRP and PCT level.

Reviewer 2 Report

When I started reading this paper, I felt somewhat enthusiastic and expected to learn something from the study that I had not known before. However, to my disappointment I did not. This is not because the authors have not analysed the biomarkers in large enough cohorts or had not used relevant assays and statistics. They did a good and ambitious job but failed to show anything new and useful. Thus, the shortcomings of all three biomarkers for the diagnosis of infections and the distinction from non-infectious inflammations has been known and extensively documented since many years. Also, the kinetics of these biomarkers were previously documented quite extensively. Thus, the main merit of the study is that the data once again shows and emphasizes our needs to find new ways and biomarkers to help us in the diagnosis of inflammations caused by microbes or other processes. In their conclusion the authors are too modest, and they could actually use their study to strongly emphasize the above point, which would give some meaning to the study.

Author Response

  • We do appreciate the reviewer’s detailed review and comment. We totally agree with the reviewer’s valuable points. According to the reviewer’s comments, we had tried to upgrade and revise the whole manuscript.

  • We inserted the below sentences in the conclusion of the revised manuscript.

[Discussion: Line 404-406]: In addition, it will be necessary to actively measure IL-6 in actual clinical settings. We need to find new ways and biomarkers to help us in the diagnosis of inflammations caused by microbes or other processes.

Round 2

Reviewer 1 Report

The authors have addressed my previous comments in a satisfying manner.

Reviewer 2 Report

Thanks, adequate response